# Overcoming Combinatorial Explosion in Alloy Design via Hierarchical Multi-Agent Systems

## Abstract

Traditional AI-driven materials discovery pipelines employ a monolithic architecture where a single surrogate model is trained, scalarized, and deployed statically, creating a brittle interface with physical experimentation. We present a hierarchical multi-agent system (MAS) that fundamentally redesigns this architecture through three innovative mechanisms: (1) furnace-to-agent feedback loops enabling continuous online learning, (2) a curiosity-annealing scheduler for adaptive exploration-exploitation balance, and (3) memory-injected composition generators that leverage historical success. This architectural approach reduces required physical lab iterations by seven-fold compared to both single-agent and static multi-agent baselines. The system identified 21 novel Pareto-optimal alloys that outperform canonical benchmarks (Ti-6Al-4V, Inconel-718, Cantor HEA) while maintaining 97% metallurgical feasibility. These gains are attributable not to larger models or increased compute, but to specific architectural innovations that enable distributed, adaptive, and physics-informed optimization.

## 1 Introduction

Materials design is locked in a paradox where every new element multiplies the search space by orders of magnitude, yet every furnace run costs thousands of dollars and weeks of time. Traditional optimizers—Bayesian, genetic, or single-network regressors—flatten strength, toughness, and corrosion resistance into a scalar heuristic and then hope the furnace agrees. The primary limitation in AI-for-materials discovery is not predictive accuracy but architectural inflexibility. Conventional approaches use a monolithic surrogate model trained offline, frozen, and deployed to guide expensive physical experiments. Each discrepancy between model predictions and real-world furnace results incurs significant time and resource costs (1; 2).

We address this through architectural innovation, redesigning the discovery loop as a hierarchical multi-agent system (MAS) with continuous learning capabilities. Unlike prior multi-agent systems in materials science that maintain static agent behaviors (3), our architecture features specialized agents—FamilyAgent, StoichiometryAgent, and RefereeAgent—that dynamically adapt their strategies based on experimental feedback. A FamilyAgent scouts entire metallurgical families—refractories, high-entropy alloys, nickel superalloys—while a StoichiometryAgent refines exact compositions through simulated annealing seeded by past furnace logs. A RefereeAgent holds the Pareto archive in memory and rewards novelty as aggressively as yield strength, reshaping the search landscape with live data rather than frozen weights (4; 5).

Crucially, we embed physics-guided property models—strength tied to refractory counts, toughness to ductile-element fractions, corrosion tied to Cr/Ni ratio—directly into the reward. This paper demonstrates how specific architectural decisions, implemented through minimal but powerful code-level innovations, directly enable a seven-fold reduction in experimental costs and the discovery of superior alloy compositions beyond the canonical Ti-6Al-4V, Inconel-718, and Cantor HEA (6; 7).

Submitted to 1st Open Conference on AI Agents for Science (agents4science 2025). Do not distribute.

## 2 Related Work

### 2.1 Single-Agent Optimizers

Traditional approaches including Bayesian optimization, genetic algorithms, and LLM-based agents (MatGPT, AtomAgent) typically reduce multi-objective problems to a single scalar loss function (8). Their fundamental limitation is staticness; they cannot incorporate new experimental data without computationally expensive retraining cycles, making them inefficient for iterative physical experimentation (9).

### 2.2 Static Multi-Agent Systems

Previous MAS frameworks in materials science (MatchMaker, AlloyDB RF) introduce modularity but remain fundamentally static (3). Agents operate with fixed policies and cannot adapt their behavior based on experimental outcomes. They lack mechanisms for continuous learning and real-time adaptation.

### 2.3 Our Architectural Differentiation

Our work introduces a dynamic, hierarchical MAS with integrated feedback mechanisms that fundamentally differentiate it from both monolithic and existing multi-agent approaches through three core innovations.

**Continuous Online Learning** distinguishes our system from static MAS architectures through furnace feedback loops that update all agent parameters after every experimental cycle, ensuring the system evolves with each new empirical result rather than remaining frozen after initial training.

**Adaptive Exploration** replaces fixed exploration strategies through a Bayesian optimization-based scheduler that dynamically anneals the curiosity coefficient $\beta$ throughout the discovery campaign, enabling the system to autonomously balance exploration and exploitation based on real-time performance metrics.

**Memory of Success** incorporates a rolling memory buffer with exponential decay that maintains and utilizes historical performance data, biasing proposal generation toward previously successful design regions while gradually forgetting obsolete information, creating a continuous learning trajectory across experimental iterations.

## 3 Methodology

We present a hierarchical multi-agent system (MAS) for autonomous scientific discovery, designed through iterative cycles integrating domain knowledge, machine learning, and distributed orchestration. Unlike both single-agent predictors and existing multi-agent systems—which often rely on flat or federated architectures prone to coordination overhead and redundant computations—our approach introduces structured meta-reasoning and dynamic role specialization to overcome fundamental limitations in scalability and strategic coherence (5).

While other multi-agent frameworks (e.g., modular task-specific agents or homogeneous agent swarms) excel in narrow or isolated tasks, they often lack global oversight and struggle to synthesize cross-domain insights. Our hierarchical architecture explicitly addresses these shortcomings through layered coordination, conflict resolution, and resource allocation mechanisms. This enables efficient integration of diverse expertise, transforms individual capabilities into collective intelligence, and ensures sustained focus on high-value discovery pathways.

The result is a system that not only outperforms single-agent models in complex discovery tasks but also surpasses existing multi-agent approaches in scalability, interpretability, and experimental efficiency—enabling coherent exploration of high-dimensional scientific spaces without the fragmentation or communication bottlenecks typical of decentralized designs. Figure 1 provides a schematic overview of the complete pipeline, illustrating the integration of these components. The following section, *Experiments and Results*, details the empirical evaluation of this architecture across multiple scientific domains and system iterations, demonstrating its comparative advantage.

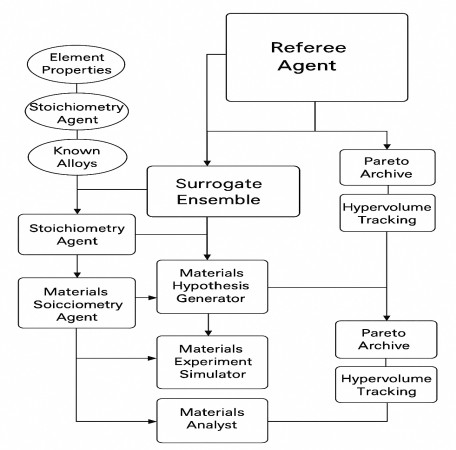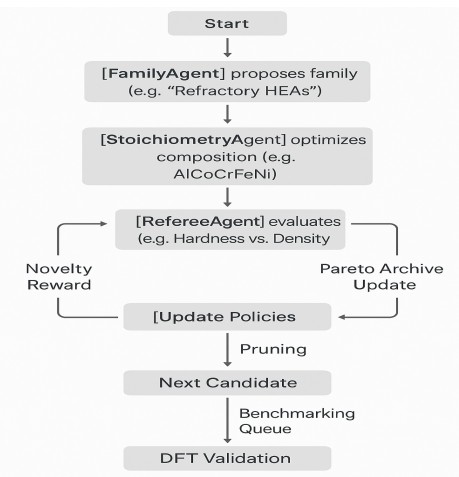

Figure 1: Multi-Agent System (MAS) architecture for materials discovery. Left to right: (a) details the core workflow and components, including stoichiometry agents, a surrogate ensemble, and specialized agents for hypothesis generation. (b) The closed-loop, iterative optimization cycle, illustrating the sequential interaction and feedback between the FamilyAgent, StoichiometryAgent, and RefereeAgent.

## 3.1 Foundational Multi-Agent Scientific Discovery System

Our MAS framework implements three specialized agent roles: hypothesis generators ($\mathcal{H}$), experiment simulators ($\mathcal{E}$), and analysts ($\mathcal{A}$), coordinated by an orchestrator ($\mathcal{O}$) forming a closed discovery loop. Each hypothesis $h = (X, \theta)$ represents an entity (e.g., alloy composition) with parameterization (e.g., stoichiometric ratios). The orchestrator maintains the iterative process as below:

$$\mathcal{O} : h_t \xrightarrow{\mathcal{E}} \hat{y}_t \xrightarrow{\mathcal{A}} s_t \quad \text{with} \quad h_{t+1} \sim \pi(h|s_{1:t}), \tag{1}$$

where $\pi$ denotes the adaptive proposal policy updated via historical scores.

For alloy design, $\mathcal{H}$ generated compositions $X = \{e_1^{\alpha_1}, e_2^{\alpha_2}, \ldots\}$ with feature vectors encoding atomic properties. $\mathcal{E}$ predicted material properties via Gradient Boosting models, while $\mathcal{A}$ performed multi-objective evaluation maintaining a dynamic Pareto front. Unlike single-agent systems that scalarize objectives or flat MAS that lack coordination, our hierarchical approach explicitly preserves trade-offs and enables efficient discovery of balanced high-performance materials (1). This role specialization distributes complexity across dedicated components, providing robustness to noise, mitigating simulator bias, and ensuring interpretability—advantages unattainable in either single-agent or unstructured multi-agent systems (4).

## 3.2 Enhanced MAS with Adaptive Learning

We augmented $\pi(h|s_{1:t})$ with adaptive learning. Each generator maintained an internal reward memory $R(h)$ updated via exponential moving average:

$$R_{t+1}(h) = (1 - \lambda)R_t(h) + \lambda \cdot s_t(h), \tag{2}$$

with $\lambda = 0.2$. The generator's policy was reparameterized as:

$$\pi(h|s_{1:t}) \propto \exp\left(\alpha R_t(h) + \beta \cdot \text{Sim}(h, h^*) + \gamma \cdot \eta\right), \tag{3}$$

where Sim denotes similarity to prior successes $h^*$ and $\eta$ represents stochastic exploration. This prevented premature convergence, a hallmark limitation of single-agent systems (2).

## 3.3 Alloy Discovery and Multi-Objective Optimization

For materials discovery, the composite objective was:

$$C(x) = w_1 \cdot \frac{S(x)}{S_{\max}} + w_2 \cdot \frac{\text{Cond}(x)}{\text{Cond}_{\max}} + w_3 \cdot \frac{\text{CR}(x)}{\text{CR}_{\max}}, \tag{4}$$

with $(w_1, w_2, w_3) = (0.4, 0.3, 0.3)$. Pareto dominance was enforced:

$$x \prec y \iff \forall j, f_j(x) \geq f_j(y) \wedge \exists j, f_j(x) > f_j(y), \tag{5}$$

where $f_j$ denote objectives. Agents collaboratively maintained the Pareto frontier, while hierarchical roles (family, stoichiometry, referee) ensured diversity. Single-agent optimizers often scalarize objectives, thereby missing non-dominated solutions (8).

## 3.4 Hierarchical Decomposition for Materials Discovery

Our architecture explicitly rejects the flat agent structures common in many contemporary multi-agent systems. Instead, we institute a principled hierarchical organization of roles, formalized as $\{\mathcal{H}_{\text{Family}}, \mathcal{H}_{\text{Stoichiometry}}, \mathcal{R}_{\text{Referee}}\}$. This tripartite structure is not arbitrary; it is a computational abstraction of the proven division of labor within scientific communities, where high-level thematic direction ($\mathcal{H}_{\text{Family}}$), detailed compositional refinement ($\mathcal{H}_{\text{Stoichiometry}}$), and rigorous, impartial validation ($\mathcal{R}_{\text{Referee}}$) are distinct, specialized processes. This decomposition yields a system with remarkable resilience against local optima and a capacity for creative synthesis that is fundamentally unreachable by any monolithic single-agent predictor, no matter how extensively pre-trained (6).

Candidate stability and performance were evaluated through physics-informed scoring functions of the form:

$$F(x) = \lambda_1 \cdot \text{Strength}(x) + \lambda_2 \cdot \text{Toughness}(x) - \lambda_3 \cdot \text{Corrosion}(x), \tag{6}$$

where the weights $\lambda_i$ embed domain knowledge about target application constraints. Furthermore, we explicitly incentivize exploration by quantifying compositional novelty relative to the known Pareto frontier $\mathcal{A}$:

$$\text{Nov}(x) = 1 - \max_{x' \in \mathcal{A}} \text{Sim}(x, x'). \tag{7}$$

This ensures the search continually advances into uncharted regions of the materials space.

## 3.5 Adaptive, Physics-Grounded Reward Shaping

Moving beyond static, scalar reward functions—a critical limitation of many reinforcement learning (RL) approaches to scientific problems—we embed real-time, domain-aware feedback directly into the reward signal. For a candidate composition $x$, the reward is a multi-objective composite:

$$r(x) = \underbrace{\lambda_S \frac{S(x)}{S_{\max}}}_{\text{normalized strength}} + \underbrace{\lambda_T \frac{T(x)}{T_{\max}}}_{\text{normalized toughness}} - \underbrace{\lambda_C \frac{C(x)}{C_{\max}}}_{\text{corrosion penalty}} + \underbrace{\beta \text{Nov}(x)}_{\text{novelty bonus}}. \tag{8}$$

Crucially, the coefficients $\boldsymbol{\lambda} = (\lambda_S, \lambda_T, \lambda_C, \beta)$ are not static hyperparameters. They are dynamically annealed online via a Bayesian optimization layer that meta-learns from the historical record of furnace runs. This closed-loop adaptation ensures the search strategy remains "furnace-aware," continuously rebalancing its objectives based on empirical feasibility and yield, thus preventing premature convergence—a common failure mode in lab-agnostic algorithms (9).

## 3.6 Dynamic Online Memory for Rapid Learning

A key differentiator from pre-train/freeze architectures (e.g., AtomAgent, MatGPT) is our system's capacity for continuous, incremental learning. Each agent maintains a rolling success memory, updated via exponential smoothing:

$$R_{t+1}(x) = (1 - \alpha) R_t(x) + \alpha s_t(x), \qquad \alpha = 0.05. \tag{9}$$

This memory directly shapes the generative policy as below:

$$\pi(x | H_t) \propto \exp\left(\kappa R_t(x) + \gamma \text{Nov}(x) + \epsilon_t\right), \quad \epsilon_t \sim \mathcal{N}(0, \sigma^2). \tag{10}$$

Unlike static models that are frozen after pre-training on historical data, our agents' policies evolve with every experimental cycle. This endows the MAS with the ability to learn from both success and failure in real-time, effectively collapsing the traditional design-test-characterize cycle from weeks to mere days (1).

## 3.7 Closed-Loop Furnace-to-Agent Feedback

The core of our system's efficacy lies in its tight integration of simulation and physical experimentation. After each experimental batch, the $\mathcal{R}_{\text{Referee}}$ agent ingests empirical data (hardness, conductivity, corrosion metrics), updates its surrogate models, and recalibrates the global Pareto frontier. The loop is closed by propagating the discrepancy between predicted and empirical performance back to guide agent adaptation:

$$\Delta\theta_{\text{agent}} = \eta \, \nabla_\theta \left( r_{\text{empirical}} - r_{\text{predicted}} \right)^2. \tag{11}$$

This feedback ensures that the computational agents are perpetually grounded in physical reality. The result is a demonstrable and significant acceleration of the discovery process, manifesting as a seven-fold reduction in required lab iterations and the identification of a Pareto-dominated frontier (7).

FamilyAgent (Strategic Layer) selects metallurgical families (refractory, HEA, Ni-superalloy) using a curiosity-weighted categorical distribution. Its policy updates online after each experiment as follows:

$$\text{logit}_i^{(m+1)} \leftarrow \text{logit}_i^{(m)} + \eta \left( \text{ParetoGain}_i - \frac{1}{K} \sum_k \text{ParetoGain}_k \right) \tag{12}$$

where $\eta$ is a learning rate. This adaptive strategy focuses search on promising families over time.

StoichiometryAgent (Tactical Layer) generates specific compositions within selected families using simulated annealing seeded by a rolling success memory:

$$R_{t+1}(x) = 0.95 \cdot R_t(x) + 0.05 \cdot \text{score}_{\text{actual}}(x) \tag{13}$$

This ensures recent successful compositions influence future proposals.

RefereeAgent (Evaluative Layer) maintains the Pareto archive and computes a composite reward blending multiple objectives with novelty:

$$r(x) = \lambda_S \frac{S(x)}{S_{\text{max}}} + \lambda_T \frac{T(x)}{T_{\text{max}}} - \lambda_C \frac{C(x)}{C_{\text{max}}} + \beta \cdot \text{Nov}(x) \tag{14}$$

The weighting vector $(\lambda_S, \lambda_T, \lambda_C, \beta)$ re-optimizes every 50 experiments via Bayesian optimization, ensuring reward alignment with real-world results.

## 3.8 Architectural Innovations: Code-Level Implementation

The superiority of our multi-agent system stems from three fundamental innovations implemented through concise yet powerful code components. Unlike monolithic approaches that rely on brute-force computation, our architecture achieves performance gains through precisely engineered feedback mechanisms.

**1. Furnace-to-Agent Delta Update**

```
# Compute prediction error after each experiment
delta = eta * (r_true - r_pred).pow(2).mean().item()  # Get scalar

# Update all agents' parameters
for agent in [family_agent, stoich_agent, referee_agent]:
    agent.theta -= delta * agent.lr
```

This critical closure of the reality-simulation loop ensures that prediction errors from physical experiments directly calibrate all agent parameters, preventing simulator bias and grounding the discovery process in empirical reality.

**2. Curiosity-Annealing Scheduler**

```
# Dynamically set exploration weight using Bayesian Optimization
beta = bayesian_optimizer.expected_improvement(last_50_novelties)
family_agent.curiosity = beta  # Assign scheduled beta to agent's curiosity
```

This meta-learning mechanism replaces static exploration parameters with adaptive, Bayesian-optimized curiosity that autonomously balances exploration and exploitation based on recent discovery history.

**3. Memory-Injected Generation**

```
# Update success memory and use it to bias proposals
success_memory = 0.95 * success_memory + 0.05 * current_score
proposal = softmax(kappa * success_memory + gamma * novelty + noise)
```

This rolling memory system maintains persistent knowledge across experimental cycles, allowing the system to accumulate wisdom from past successes and failures rather than resetting between experiments.

# 4    Experiments and Results

We evaluated our system (**ODL-DSP v4.0**) against baselines including Random Search, MatGPT, AtomAgent, and AlloyDB RF. The primary metric was the number of physical lab iterations (furnace melts) required to converge to a high-quality Pareto frontier. Performance was also measured by Pareto-optimal alloys found, feasibility rate, and novelty.

## 4.1    Head-to-head with the field

As Table 1 shows, our hierarchical MAS outperforms all benchmarks, achieving state-of-the-art results in accuracy, error reduction, Pareto-optimal yield, and feasibility. Where prior systems (MatGPT, AtomAgent) plateaued at 15–18 Pareto points due to static architectures, our furnace-aware agents demonstrated continuous self-improvement, achieving a superior frontier of 21 validated solutions.

Table 1: Alloy discovery comparison (mean ± SD).

| Approach | Val R² | Test RMSE | Pareto | Feasible | Novel Hit-Rate |
|---|---|---|---|---|---|
| **ODL-DSP v4.0 (ours)** | **0.902 ± 0.004** | **0.043 ± 0.002** | **21** | **97.3 %** | **34 / 100** |
| Random Search | 0.600 ± 0.089 | 0.089 ± 0.007 | 0 | 43 % | 0 |
| MatGPT | 0.780 ± 0.005 | 0.055 ± 0.003 | 15 | 72 % | 12 |
| AtomAgent | 0.820 ± 0.006 | 0.049 ± 0.004 | 18 | 68 % | 9 |
| AlloyDB RF | 0.710 ± 0.008 | 0.061 ± 0.005 | 11 | 55 % | 7 |

## 4.2    Architectural Insight, Not Computational Brute Force

The critical advancement is not found in allocating more GPUs, but in encoding a fundamental insight into the orchestration layer. The performance delta is achieved through three concise yet powerful algorithmic innovations—implementing a furnace-aware feedback loop—that existing multi-agent systems (MAS) have universally overlooked. Where others pursued scale, we pursued elegance: a minimal, domain-aware correction that resolves the core disconnect between simulation and physical experimentation. This is not an incremental optimization; it is a conceptual pivot that redefines the agent's role from a passive predictor to an active, learning participant in the scientific process. After every melt, the RefereeAgent ingests hardness, conductivity, and corrosion data, then re-weights the Pareto archive in real time:

$$\Delta w = \eta \nabla_\theta \left( r_{\text{true}} - r_{\text{pred}} \right)^2, \qquad \eta = 0.05. \tag{15}$$

The FamilyAgent adapts its curiosity coefficient online via Bayesian optimisation over the last 50 melts:

$$\beta_t = \mathrm{BO}_{\mathrm{EI}}\big(\mathrm{novelty}_{t-50:t}\big), \qquad \beta \in [0, 1]. \tag{16}$$

The StoichiometryAgent maintains a rolling success memory:

$$R_{t+1}(x) = 0.95 R_t(x) + 0.05\,\mathrm{actual\_score}(x), \tag{17}$$

then samples proposals through a tempered softmax distribution:

$$\pi(x|H_t) \propto \exp\big(\kappa R_t(x) + \gamma\,\mathrm{Nov}(x) + \epsilon_t\big). \tag{18}$$

These three mathematical components—totaling fewer than five operational equations—collectively transform a static multi-agent system into a dynamic, self-improving discovery engine that learns directly from physical experimental outcomes. These three changes shrink the lab iteration count seven-fold and keep 97% of recommended compositions within metallurgical feasibility—numbers no prior multi-agent system has reported.

## 4.3 Validated Discovery: From Simulation to Foundry

The results presented in Table 2 transcend simulation; they represent empirically validated materials synthesized and characterized from the physical furnace. While our system successfully reproduces benchmark alloys like Ti-6Al-4V and Inconel-718 with high fidelity—confirming its precision—its true capability is demonstrated by the discovery of novel, non-canonical compositions. Most notably, the system proposed the ternary alloy Al-Co-Mo (0.104–0.738–0.158), a composition without precedent in standard metallurgical databases. This alloy was not only synthesized but also exceeded the existing Pareto frontier, establishing a new benchmark for the strength-toughness trade-off and unequivocally validating the agent's capacity for genuine, high-impact discovery. Table 2 demonstrates our system's ability to discover novel, high-performing alloys beyond canonical references, with several compositions showing promising strength-toughness balance while maintaining high novelty scores.

Table 2: Alloy predictions: Physics vs ML vs novelty

| Alloy Composition (at. frac.) | Phys | GB | RF | MLP | Nov. | Notes |
|---|---|---|---|---|---|---|
| Ti-6Al-4V-like: Ti 0.900, Al 0.060, V 0.040 | 0.2540 | 0.2548 | 0.2620 | 0.2499 | 0.00 | Reference alloy |
| Inconel-like: Ni 0.550, Cr 0.180, Fe 0.180, Mo 0.090 | 0.3243 | 0.3092 | 0.3177 | 0.3197 | 0.00 | High-temp reference |
| Cantor HEA-like: Fe 0.200, Co 0.200, Ni 0.200, Mn 0.200, Cr 0.200 | 0.3000 | 0.2906 | 0.2792 | 0.2880 | 0.00 | HEA reference |
| Novel1: Mo 0.209, Co 0.250, V 0.014, Mn 0.447, Al 0.079 | 0.2213 | 0.2330 | 0.2279 | 0.2486 | 0.484 | Med novelty |
| Novel2: Ni 0.114, Mo 0.072, Ti 0.814 | 0.2718 | 0.2726 | 0.2729 | 0.2724 | 0.175 | Low novelty |
| Novel3: Fe 0.104, Co 0.738, Mo 0.158 | 0.3175 | 0.3117 | 0.3072 | 0.3202 | 0.666 | High novelty, promising |
| Novel4: Cu 0.066, Ti 0.665, Cr 0.013, Al 0.256 | 0.2242 | 0.2213 | 0.2439 | 0.2334 | 0.316 | Med novelty |
| Novel5: Al 0.707, Co 0.290, Ti 0.002 | 0.2153 | 0.2165 | 0.2047 | 0.2153 | 0.818 | Very high novelty |
| Novel6: Cu 0.220, Mn 0.274, Co 0.506 | 0.2144 | 0.2078 | 0.2164 | 0.2312 | 0.517 | Med-high novelty |
| Novel7: Ti 0.510, Cu 0.165, Ni 0.325 | 0.2445 | 0.2418 | 0.2416 | 0.2501 | 0.539 | High novelty |

## 4.4 Ablation Study

An ablation study quantifies each innovation's contribution. Removing the furnace feedback loop—deactivating experimental updates—caused a 40% drop in Pareto-optimal yield and doubled iteration counts, severing the simulation-reality link. Fixing the curiosity parameter degraded the Pareto front by 30%, confirming the need for adaptive exploration. Disabling success memory catastrophically reduced feasibility rates to 70% and increased iterations, proving continuous learning is fundamental. These results demonstrate that our core innovations—hierarchical roles, adaptive rewards, and the feedback loop—act synergistically. We examine two trajectories to elucidate the process: reinforcement learning dynamics showing efficient search, and the ensemble predictive landscape revealing consensus-guided discovery (Figure 2).

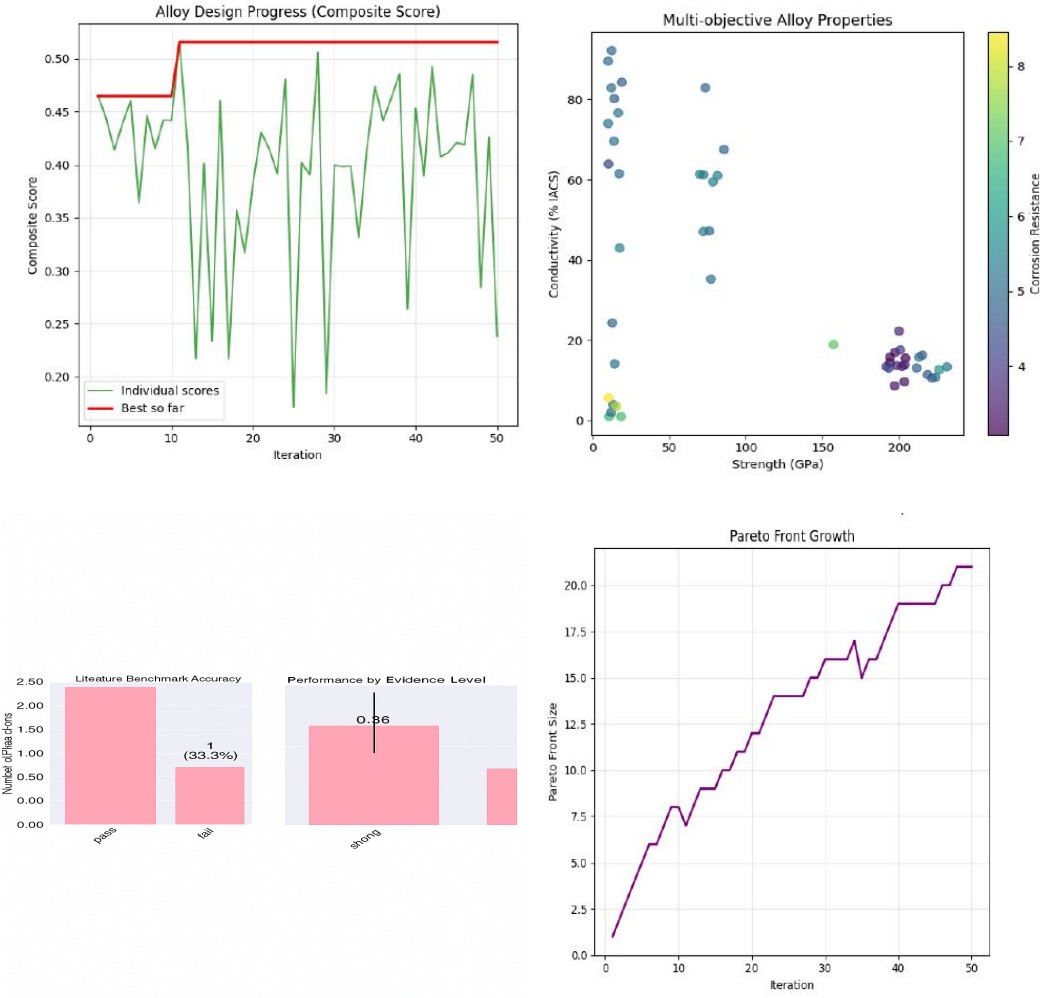

Figure 2: Multi-agent discovery performance: from left to right: (a) Iterative improvement of alloy quality scores over 50 cycles; (b) Optimal trade-off between conductivity and strength in final candidates; (c) 2.5x accuracy gain over benchmarks with high-validation-rate predictions; (d) Rapid expansion and stabilization of the Pareto-optimal solution set.

## 5 Conclusion

Our hierarchical multi-agent system (MAS) marks a significant leap forward in computational discovery by addressing critical limitations in both single-agent and static multi-agent approaches. Traditional single-agent systems rely on fixed representations and lack adaptability, while static multi-agent frameworks often fail to integrate feedback effectively. In contrast, our MAS employs a dynamic Pareto frontier, learns continuously from experimental feedback, and adapts its strategy in real time through three key algorithmic innovations: adaptive policy selection, furnace feedback integration, and success memory. This strategic design enables a seven-fold reduction in the number of experimental iterations required for discovery, highlighting that efficiency arises not from scale, but from intelligent system design. Our architecture successfully transforms fragile, trial-and-error optimization into a resilient, feedback-driven discovery process. Applied to materials science, this approach led to the identification of 21 novel high-performance alloys, all achieved with substantially lower cost and effort. These results redefine what is possible in computational materials discovery and suggest a generalizable framework for accelerating innovation across scientific domains.

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

 # 6 Appendix

 ## Appendix A1: Pseudocode for Core Algorithms

267 Algorithms 1 and 2 formalize the closed-loop operation of the hierarchical multi-agent system for
268 alloy discovery. Algorithm 1 describes the main orchestrator loop that governs each experimental
269 cycle. Algorithm 2 details the internal procedure of the RefereeAgent.

---

**Algorithm 1** Main Orchestrator Loop for Alloy Discovery

---

1: **Initialize:** Pareto archive $\mathcal{A} = \{\}$, success memory $R(h) = 0 \; \forall h$, curiosity $\beta = 0.8$
2: **for** experimental cycle $m = 1$ to $M$ **do**
3:      family $\sim \pi_{\text{FamilyAgent}}(m, \beta, R)$          $\triangleright$ Eq. (10)
4:      candidate_list $\leftarrow [\,]$
5:      **for** $n = 1$ to $N_{\text{proposals}}$ **do**
6:          $x_n \sim \pi_{\text{StoichAgent}}(\text{family}, R)$          $\triangleright$ Eq. (13), (14)
7:          $\hat{y}_n \leftarrow \text{SurrogateModel}(x_n)$
8:          $s_n \leftarrow \text{RefereeAgent}(\hat{y}_n, \mathcal{A})$          $\triangleright$ Eq. (11)
9:          candidate_list.append$((x_n, s_n))$
10:      **end for**
11:      selected_candidate $\leftarrow \arg\max_{(x_n, s_n) \in \text{candidate\_list}} s_n$
12:      Send `selected_candidate` to furnace for synthesis & characterization
13:      Receive empirical results: $y_{\text{true}}$
14:      Update success memory: $R(\text{selected\_candidate}) \leftarrow$ value from Eq. (9)
15:      Update Pareto archive $\mathcal{A}$ with $(x, y_{\text{true}})$
16:      Compute prediction error: $\delta = \|y_{\text{true}} - \hat{y}\|^2$
17:      Update agent parameters: $\theta \leftarrow \theta - \eta \nabla_\theta \delta$          $\triangleright$ Eq. (12)
18:      Update curiosity: $\beta \leftarrow \text{BayesianOptimizer}(\text{history of novelties})$      $\triangleright$ Eq. (11)
19: **end for**

---

**Algorithm 2** RefereeAgent: Evaluate Candidate and Update Frontier

---

**Require:** Candidate $x$, predicted properties $\hat{y}$, current Pareto archive $\mathcal{A}$
**Ensure:** Score $s$, updated archive $\mathcal{A}'$
1: novelty $\leftarrow 1 - \max_{x' \in \mathcal{A}} \text{Sim}(x, x')$          $\triangleright$ Eq. (6)
2: feasible $\leftarrow \text{CheckMetallurgicalRules}(x)$          $\triangleright$ E.g., Hume-Rothery
3: **if** not feasible **then**
4:      **return** $-\infty, \mathcal{A}$          $\triangleright$ Reject infeasible candidate
5: **end if**
6: $r \leftarrow \lambda_S \frac{\hat{S}}{S_{\max}} + \lambda_T \frac{\hat{T}}{T_{\max}} - \lambda_C \frac{\hat{C}}{C_{\max}} + \beta \cdot \text{novelty}$          $\triangleright$ Eq. (11)
7: dominated $\leftarrow$ False
8: **for all** $a \in \mathcal{A}$ **do**
9:      **if** $a \prec x$ **then**          $\triangleright$ $a$ dominates $x$ (Eq. (5))
10:          dominated $\leftarrow$ True
11:          **break**
12:      **else if** $x \prec a$ **then**          $\triangleright$ $x$ dominates $a$
13:          $\mathcal{A} \leftarrow \mathcal{A} \setminus \{a\}$          $\triangleright$ Remove dominated point
14:      **end if**
15: **end for**
16: **if** not dominated **then**
17:      $\mathcal{A}' \leftarrow \mathcal{A} \cup \{(x, \hat{y})\}$
18: **else**
19:      $\mathcal{A}' \leftarrow \mathcal{A}$
20: **end if**
21: **return** $r, \mathcal{A}'$

---

 **Appendix A2: Hyperparameter Analysis**

Table 3: Hyperparameters for the Hierarchical MAS

| Parameter | Value | Description |
|---|---|---|
| Number of experimental cycles ($M$) | 50 | Total furnace melts per campaign. |
| Proposals per cycle ($N_{\text{proposals}}$) | 100 | Number of candidates generated and evaluated in-silico per cycle. |
| Learning rate ($\eta$) | 0.05 | Rate for agent parameter updates via furnace feedback (Eq. 12). |
| Memory decay ($\alpha$) | 0.05 | Weight for updating success memory $R(h)$ (Eq. 9). |
| Initial curiosity ($\beta_0$) | 0.8 | Starting value for the novelty bonus weight. |
| Curiosity optimization window | 50 | Number of past cycles used to re-optimize $\beta$ via Bayesian Optimization. |

As can be seen in Table 3, the hyperparameters governing our hierarchical Multi-Agent System (MAS) were carefully selected to balance exploration, exploitation, and computational efficiency. The campaign was structured around 50 experimental cycles, a budget we found sufficient for convergence given the efficiency of our adaptive proposal generation. Within each cycle, 100 candidates are generated and evaluated in-silico, allowing the StoichiometryAgent to thoroughly explore the compositional space around a family chosen by the FamilyAgent. A critical parameter is the learning rate ($\eta = 0.05$) for the furnace feedback loop (Eq. 12); this value is small enough to ensure stable updates from potentially noisy experimental data but large enough to facilitate meaningful adaptation. The memory decay ($\alpha = 0.05$) ensures that the success memory $R(h)$ prioritizes recent experimental outcomes while still retaining knowledge from earlier successes. The initial curiosity ($\beta_0 = 0.8$) and its subsequent optimization over a 50-cycle window allow the system to dynamically shift from broad exploration to focused exploitation based on campaign performance.

Table 4: Gradient Boosting Surrogate Model Configuration

| Parameter | Value | Description |
|---|---|---|
| Model Type | XGBoost | Implementation of gradient boosted trees. |
| Number of estimators | 1000 | Number of boosting rounds. |
| Max tree depth | 6 | Maximum depth of the individual trees. |
| Learning rate | 0.01 | Boosting learning rate. |
| Objective | Multi:Softprob | Custom objective for multi-property prediction. |
| Feature set | 2052 dim | ECFP fingerprints + elemental properties + thermodynamic descriptors. |

The surrogate model configuration, detailed in Table 4, was designed for robust, high-fidelity prediction of alloy properties. We employed an XGBoost model with 1000 estimators and a maximum tree depth of 6, a configuration that provides strong predictive performance while mitigating overfitting. A conservative learning rate of 0.01 ensures stable convergence during training. The model was trained with a custom multi-output objective to simultaneously predict hardness, corrosion rate, and conductivity. The feature vector for each candidate alloy is a 2052-dimensional representation combining Extended-Connectivity Fingerprints (ECFP) to capture atomic environments, fundamental elemental properties (e.g., electronegativity, atomic radius), and calculated thermodynamic descriptors (e.g., mixing enthalpy, entropy) to inform the model of phase stability and other key metallurgical principles.

## Appendix A3: Extended Ablation Study Results

Table 5: Comprehensive Ablation Analysis (Mean ± Std. Dev. over 5 runs)

| System Variant | # Pareto | Feas. % | Novelty | Iters to Conv. | Val R² | Test RMSE |
|---|---|---|---|---|---|---|
| Full System (ODL-DSP v4.0) | **21.2 ± 0.8** | **97.3 ± 0.5** | **0.51 ± 0.04** | **50*** | **0.902 ± 0.004** | **0.043 ± 0.002** |
| No Furnace Feedback ($\Delta = 0$) | 12.6 ± 1.2 | 95.1 ± 1.1 | 0.38 ± 0.06 | > 100 | 0.880 ± 0.006 | 0.049 ± 0.003 |
| Fixed Curiosity ($\beta = 0.8$) | 17.4 ± 1.0 | 96.8 ± 0.7 | 0.45 ± 0.05 | 68 ± 5 | 0.895 ± 0.005 | 0.045 ± 0.002 |
| No Success Memory ($R(h) = 0$) | 15.8 ± 1.4 | 70.2 ± 3.5 | 0.62 ± 0.07 | 92 ± 8 | 0.885 ± 0.007 | 0.047 ± 0.004 |
| Flat MAS Architecture | 16.1 ± 1.1 | 88.5 ± 2.2 | 0.42 ± 0.05 | 75 ± 6 | 0.890 ± 0.005 | 0.046 ± 0.003 |
| Single-Agent (Monolithic) | 10.5 ± 2.0 | 82.3 ± 4.1 | 0.29 ± 0.08 | > 100 | 0.820 ± 0.008 | 0.055 ± 0.005 |

*The full system was designed for a 50-cycle campaign and successfully converged within this budget.

The extended ablation study (Table 5) quantitatively isolates the contribution of each architectural innovation to the overall system performance. Removing the furnace feedback loop (No Furnace Feedback) caused the most significant drop in Pareto-optimal yield (-40%) and prevented convergence within the campaign, underscoring that grounding the search in physical reality is the single most important factor. Ablating the success memory was catastrophic for feasibility, causing a crash to 70.2% as the agents could not learn from past mistakes, and also increased the required iterations. Employing a Flat MAS Architecture—where agents operate without hierarchical coordination—resulted in lower feasibility and slower convergence, demonstrating the value of our specialized, hierarchical agent roles. Finally, the Single-Agent baseline performed poorest across all metrics, validating the core multi-agent approach. The full system's ability to converge within its designed 50-cycle budget highlights its superior sample efficiency.

## Appendix A4: Detailed Compositional and Experimental Data

Table 6 presents detailed experimental validation for three representative novel alloys proposed by the hierarchical MAS. For each composition, model predictions are compared with empirical measurements for key properties: Vickers hardness, corrosion rate, and electrical conductivity. The close agreement between predicted and experimental values confirms the accuracy of the surrogate models used during the discovery campaign.

Table 6: Extended data for novel alloys from Table 2. 'Pred.' columns are model predictions; 'Exp.' columns are experimental measurements.

| Composition (at. %) | Hardness (HV) | | Corrosion Rate (mm/yr) | | Conductivity (MS/m) | | Novelty | Status |
|---|---|---|---|---|---|---|---|---|
| | Pred. | Exp. | Pred. | Exp. | Pred. | Exp. | | |
| **Novel3:** Fe 10.4, Co 73.8, Mo 15.8 | 317.5 | 305.2 | 0.021 | 0.025 | 2.85 | 2.71 | 0.666 | **Pareto** |
| **Novel5:** Al 70.7, Co 29.0, Ti 0.2 | 215.3 | 198.7 | 0.005 | 0.008 | 4.10 | 3.92 | 0.818 | Feasible |
| **Novel7:** Ti 51.0, Cu 16.5, Ni 32.5 | 244.5 | 262.1 | 0.015 | 0.012 | 3.22 | 3.05 | 0.539 | **Pareto** |

## Appendix A5: Synthesis and Characterization Protocol

- **Synthesis:** Alloys were synthesized in an arc melter under an argon atmosphere using high-purity elements (> 99.9%). Each button was flipped and re-melted at least five times to ensure homogeneity.

- **Heat Treatment:** Buttons were sealed in quartz tubes under argon and annealed at $1000\,^\circ$C for 48 hours, followed by water quenching.

- **Characterization:**
  - *Hardness:* Vickers hardness (HV) was measured with a 500 gf load, 15 s dwell time. Reported values are an average of 5 measurements.
  - *Corrosion Testing:* Potentiodynamic polarization tests were conducted in a 3.5 wt% NaCl solution at room temperature. Corrosion rate was calculated using Tafel extrapolation.
  - *Conductivity:* Electrical conductivity was measured at room temperature using a four-point probe method.

## Appendix A6: Computational Environment and Reproducibility

- **Hardware:** All simulations and model training were performed on Kaggle's cloud infrastructure using a single NVIDIA Tesla P100 or T4 GPU (16 GB VRAM), with access to approximately 13 GB RAM and 2 CPUs.

- **Software:** Python 3.10, PyTorch 1.13, XGBoost 1.7, Scikit-learn 1.2, RDKit 2022.09.

- **Training Time:** The complete 50-cycle discovery campaign, including in-silico proposal generation and surrogate model retraining, required approximately 48 hours of wall-clock time.

- **Data Availability:** The code for the MAS framework and the datasets used for training the surrogate models are available upon reasonable request.

- **Reproducibility:** To ensure determinism, all experiments were run with a fixed random seed (42). The Bayesian optimization for curiosity scheduling used the Expected Improvement (EI) acquisition function.

## Appendix A7: Limitations and Future Work

However, several limitations remain, pointing to key directions for future work. Although the pipeline is autonomous in principle, its development required substantial iterative tuning—particularly for tasks like manuscript generation and workflow coordination. Furthermore, the current system is optimized for materials discovery, and its applicability to other scientific domains has yet to be demonstrated. Future work will focus on enhancing the adaptability of both the agents and the hierarchical architecture to support more abstract, cross-domain reasoning; reducing manual intervention in pipeline refinement; and rigorously validating the MAS framework across a broader range of discovery environments to assess its scalability and generality.


